# Non-Poissonian photon statistics from macroscopic photon cutting materials

Mathijs de Jong[1], Andries Meijerink[1] & Freddy T. Rabouw[1]

In optical materials energy is usually extracted only from the lowest excited state, resulting in fundamental energy-efficiency limits such as the Shockley–Queisser limit for single-junction solar cells. Photon-cutting materials provide a way around such limits by absorbing high-energy photons and 'cutting' them into multiple low-energy excitations that can subsequently be extracted. The occurrence of photon cutting or quantum cutting has been demonstrated in a variety of materials, including semiconductor quantum dots, lanthanides and organic dyes. Here we show that photon cutting results in bunched photon emission on the timescale of the excited-state lifetime, even when observing a macroscopic number of optical centres. Our theoretical derivation matches well with experimental data on $NaLaF_4:Pr^{3+}$, a material that can cut deep-ultraviolet photons into two visible photons. This signature of photon cutting can be used to identify and characterize new photon-cutting materials unambiguously.

---

[1] Debye Institute for Nanomaterials Science, Utrecht University, Princetonplein 1, 3584 CC Utrecht, The Netherlands. Correspondence and requests for materials should be addressed to F.T.R. (email: f.t.rabouw@uu.nl).

Optical materials are used to convert the energy of photons into other useful forms. Examples include photovoltaic materials converting light into electrical energy, phosphors transforming one colour of light into another and photocatalysts using photon energy to make or break chemical bonds. In most materials the conversion process generates no more than one quantum of output energy (for example, an energetic electron or a colour-converted photon) per one photon absorbed. The energy efficiency of such processes has a fundamental limit that is approximately inversely proportional to the energy of the incoming photon. This is, for example, a major factor determining the Shockley–Queisser efficiency limit for single-junction solar cells[1].

The process of photon cutting or quantum cutting can substantially improve the energy conversion efficiency for high-energy photons by 'cutting' them into multiple lower-energy excitations. This possibility was first hypothesized in 1957 (ref. 2) and has since been demonstrated experimentally in various materials, including semiconductor quantum dots (multi-exciton generation)[3–7], organic dyes (singlet fission)[8–14] and lanthanide ions[15–18].

Most experiments for the characterization of photon-cutting materials rely on (time-resolved) photoluminescence or transient absorption measurements as a function of excitation wavelength[3,6] or density of optical centres[12,16,17]. Usually, the proof that photons are 'cut' into multiple excitations is indirect, except in rare cases where the excitations can be extracted with very high efficiency[7,19]. This leads to ambiguities in the identification and characterizations of new photon-cutting materials. For example, the occurrence of multi-exciton generation in semiconductor quantum dots is usually concluded from fast decay components in transient absorption data[3,20–22], but these can also originate from trapping of charge carriers or charging of the quantum

dots[23]. Triplet states in dye molecules can be generated by the photon-cutting process of singlet fission[11,14], but also by regular intersystem crossing[9]. Similarly, non-radiative energy transfer from a highly excited lanthanide ion can result in photon cutting through distribution of the energy over multiple excited centres[15,16], but it is not trivial to distinguish this from processes generating only one excitation, while excess energy is lost as heat[24]. In view of this, it is not surprising that previous studies have sometimes reported contradictory conclusions on the occurrence or efficiency of photon cutting[20,21,24].

Here we propose that direct proof of photon cutting in a material is possible by the observation of non-Poissonian photon emission statistics. Bunched emission has been reported from multi-exciton states in single CdSe quantum dots[25]. However, analysing photon cutting on a single optical centre is challenging at best and impossible for many photon cutters that rely on energy transfer between centres. We derive here that a photon-cutting material exhibits photon bunching even if it contains a macroscopic number of optical centres. Photon bunching should therefore be observable from any photon cutter where the excitations can be extracted as light[6,15,16,23,26,27]. We demonstrate this phenomenon experimentally on the photon-cutting phosphor NaLaF$_4$ doped with Pr$^{3+}$ (ref. 28).

## Results

**Derivation of photon bunching from a photon-cutting material.** We start from the general energy level scheme of a photon cutter (Fig. 1a). A high-energy photon (purple) excites the system to a high excited state Y. This is followed by a cascade of transitions: first to an intermediate excited state X (blue) and then further to the ground state G (red). In both steps of the cascade, a photon is emitted. Figure 1b is an exemplary photon detection trace of a single

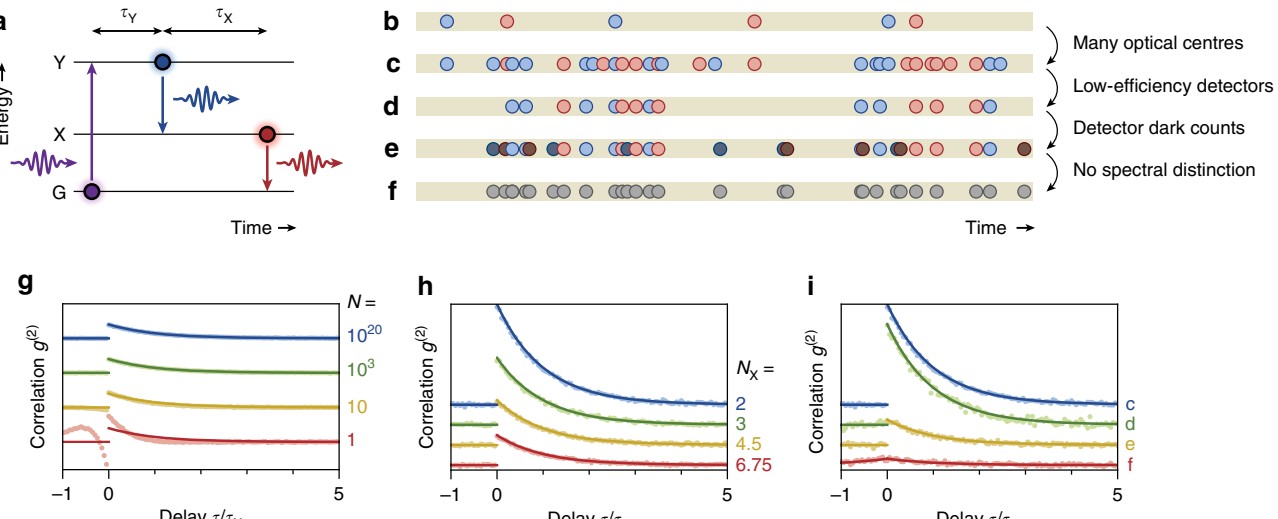

**Figure 1 | The detection of photon bunching from a macroscopic photon-cutting material.** (**a**) Photon cutting in a material with ground state G, first excited state X and second excited state Y. A high-energy photon (purple) excites the system to excited state Y, after which a cascade of two relaxations take place: from Y to X (blue) and from X to G (red). (**b**) Photon detection trace of a single ideal optical centre as in **a**, assuming ideal detectors. (**c**) Photon detection trace similar to **b**, but for an ensemble of optical centres. (**d**) Photon detection trace similar to **c**, but for a detection efficiency of 0.5. Because of imperfect detection some of the photons are missed. (**e**) Photon detection trace similar to **d**, but assuming a dark count rate equal to the signal strength. The dark-coloured photons are dark counts, which in an experiment would be indistinguishable from regular counts. (**f**) Photon detection trace as in **e**, but without spectral distinction between the photons of different colour. (**g**) Cross-correlation functions for an increasing number of optical centres ($N = 1$, 10, $10^3$ and $10^{20}$). The solid lines are the analytical function for the limit of a large ensemble, the data points the results of Monte Carlo simulations. (**h**) Cross-correlation function at different excitation intensities, which correspond to different values for the steady-state population of the intermediate state. The solid lines are the analytical solutions and the data points the results of Monte Carlo simulations. (**i**) Cross-correlation functions for the experimental scenarios sketched in **c**–**f**. The solid lines are the analytical functions, the data points the results of Monte Carlo simulations. In Supplementary Note 1 we derive the analytical correlation functions for non-ideal scenarios.

optical centre exhibiting such cascade emission. All photons from the first transition (blue) are followed by a photon from the second transition (red), with as average time interval the decay time $\tau_X$ of the intermediate state X. For a material with a large number $N$ of optical centres the emitted photons are also bunched in pairs, but now the pairs overlap in time (Fig. 1c). The statistics of bunched emission are different from the statistics of regular (that is, non-photon cutting) photoluminescence. The peculiar photon statistics of a photon cutting material can be described mathematically and investigated experimentally using the normalized photon–photon cross-correlation function.

$$g^{(2)}(\tau) = \frac{\langle I_1(t)I_2(t+\tau)\rangle}{\langle I_1(t)\rangle\langle I_2(t)\rangle}, \qquad (1)$$

where $I_1$ is the intensity of the first emission step in the cascade and $I_2$ is the intensity of the second step. The cross-correlation function describes how likely it is to detect a photon from the second transition at time $\tau$ after detection of a photon from the first transition. In the Methods section, we derive the cross-correlation function analytically for an experiment on a macroscopic photon cutter (as in Fig. 1c) in which the first photon (blue in Fig. 1a) and the second photon (red in Fig. 1a) are spectrally separated and directed to two independent detectors with negligible dark count rates:

$$g^{(2)}(\tau) = 1 + \frac{1}{\bar{N}_X}e^{-k_{XG}\tau} \qquad (2)$$

with $\bar{N}_X$ the steady-state population and $k_{XG}$ the decay rate of the intermediate level X. Bunching is observed as an additional signal decaying with rate $k_{XG}$ on top of a constant unity background caused by Poissonian photons statistics.

In Fig. 1g, we plot the analytical correlation function (lines) for an ideal photon cutting material together with the correlation function from a Monte Carlo simulation (dots), for different numbers of optical centres ($N$) in the material. The excitation rate is set at $\Phi = 2k_{XG}/N$, which corresponds to a constant steady-state population of $\bar{N}_X = 2$ in the limit of large $N$. The cross-correlation function shows an increased likelihood of detecting a (red) photon from the second transition after detection of a (blue) photon from the first transition for any number $N$ of optical centres. The analytical model (equation (2)) matches well with the Monte Carlo results for an ensemble of emitters with $N > 10$, including macroscopic materials containing a number of optical centres on the order of Avogadro's constant (Fig. 1g, green and blue), clearly revealing the occurrence of photon-pair emission in the photon statistics. The analytical model is less accurate for a small $N$ (yellow and red), because the approximation of no ground-state depletion is justified only for large $N$ and low excitation rates. In agreement with equation (2), Fig. 1h shows that an increasing excitation rate $\Phi$ (and therefore increasing steady-state population $\bar{N}_X$) results in a smaller bunching amplitude. At the same time a higher excitation rate $\Phi$ results in a higher photon count rate and therefore in decreased statistical noise on the correlation function. Interestingly, the bunching signal and the statistical noise both scale with $1/\Phi$, so that the excitation rate has no net effect on the signal-to-noise ratio in a photon-bunching experiment. See Supplementary Fig. 3 for an analysis of what this means for the fit uncertainties.

Figure 1c–f illustrate complications that may arise in real experiments on photon-cutting materials. Ideally, the collection and detection of photons would have unity efficiency (Fig. 1c), but in practice this efficiency is finite so that many photon emission events go unnoticed (Fig. 1d). Moreover, detectors have a finite dark count rate leading to random background counts (Fig. 1e). Finally, for many photon-cutting materials it is not possible to spectrally distinguish the two photons emitted in the

cascade process (Fig. 1f). The correlation functions for increasing experimental complexity are plotted in Fig. 1i. A non-unity detection efficiency does not lower the bunching amplitude, but only lowers the count rates and therefore increases the noise on the data (compare blue and green). Detector dark counts lower the bunching amplitude (yellow), and should therefore be minimal. (In Supplementary Note 1 we investigate the effect of dark counts analytically.) The bunching amplitude decreases further if the two emitted photons cannot be separated spectrally (red). The bunching signal now becomes symmetric about $\tau = 0$, because the order of detector clicks is no longer sensitive to order of the emitted photons (Fig. 3d).

**Experimental demonstration of photon cutting in NaLaF$_4$:Pr$^{3+}$.** To experimentally test the occurrence of photon bunching in the emission from a macroscopic photon-cutting material, we measure the phosphor NaLaF$_4$ doped with 1% Pr$^{3+}$ (ref. 28). We use $\sim$10 mg of the material, containing $N = 10^{17}$–$10^{18}$ optical centres that in the experiment are excited more or less homogeneously. Figure 2a shows the mechanism of photon cutting in Pr$^{3+}$: ultraviolet light excites the ion to the $^1S_0$ level (level Y in Fig. 1a), after which a radiative transition to the $^1I_6$ level, rapid non-radiative relaxation to the $^3P_0$ level (level X in Fig. 1a) and a radiative transition to one of the $^3H_J$ levels (level G in Fig. 1a) follow[29,30]. In Supplementary Fig. 2 and Supplementary Note 2, we discuss how other types of photon cutters, such as quantum dots exhibiting multiple-exciton generation, can be mapped on the model of Fig. 1a.

Our experimental setup is sketched in Fig. 2b. The blue photons from the first radiative transition and the red–green photons from the second radiative transition are separated (Fig. 2c,d) with a dichroic filter and sent to two independent detectors. Cross-correlating the two detector signals yields Fig. 2e. We observe an increased likelihood of detecting a red-green photon after detection of a blue photon. The inset of Fig. 2e shows that temporal decay of the bunching signal in the cross-correlation function (green) matches the photoluminescence decay of the intermediate $^3P_0$ level (red), in agreement with equation (2). This is a direct proof of photon cutting in NaLaF$_4$:Pr$^{3+}$, with the $^3P_0$ level of Pr$^{3+}$ as the intermediate state.

**The magnitude of photon bunching.** In Fig. 3a–c we show how the bunching signal changes with excitation density $\Phi$. The amplitude of the bunching signal (Fig. 3a) is inversely proportional to the excitation density, while the decay time is constant at 18.4 μs (Fig. 3b). This decay time is in good agreement with the reported 18 μs[28] of the intermediate state $^3P_0$ ($=$ X in Fig. 1a). Meanwhile, the ratio of count rates on the two detectors is constant at 0.33 red photons per 1 blue photon, in good agreement with the reported value of 0.40 (ref. 28).

With the aim to investigate the effect of spectral separation, we have performed the experiment both with and without spectrally separating the two photons emitted in the cascade process. Spectral separation was achieved by using a dichroic mirror in the emission path (as in Fig. 2e), while the experiment without spectral separation used a 50/50 beamsplitter. In Fig. 3d we show the two resulting cross-correlation functions. The cross-correlation function with spectral separation (green) is similar to the result in Fig. 2e. The cross-correlation function of the experiment with the 50/50 beamsplitter (grey) shows symmetric bunching with a lower amplitude, in agreement with the simulations in Fig. 1i (red). We derive in Supplementary Note 1 that the ratio between the bunching amplitudes of the experiments with and without spectral separation is related to the radiative efficiencies of the two transitions in the photon-cutting material. Based on

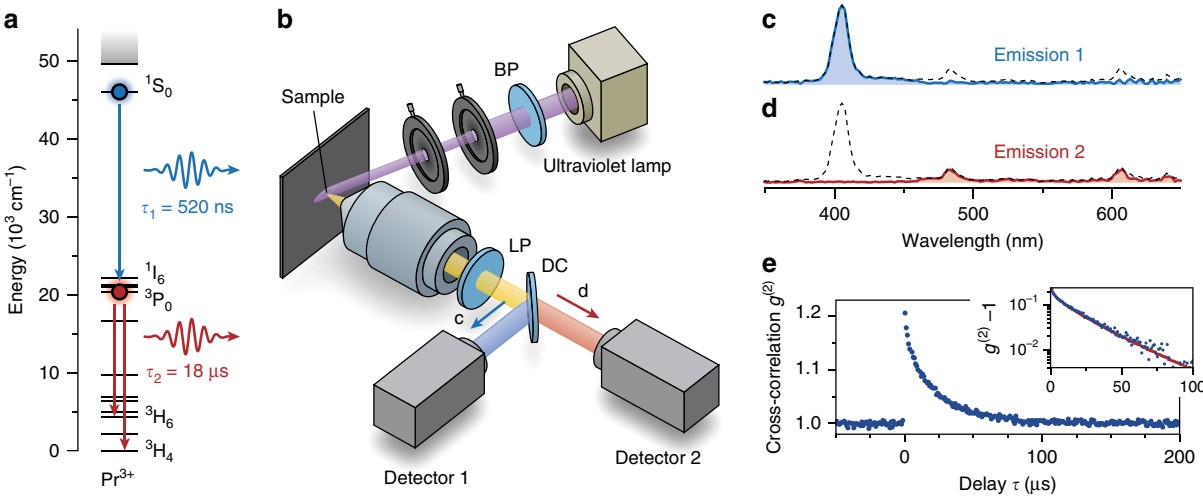

**Figure 2 | Experimental observation of photon bunching from a NaLaF$_4$:Pr$^{3+}$ powder.** (**a**) Energy level diagram of Pr$^{3+}$ in NaLaF$_4$ (ref. 28). After absorption of a ultraviolet photon, Pr$^{3+}$ ends up in the $^1S_0$ excited state. From this level, Pr$^{3+}$ decays to the $^1I_6$ level, emitting a blue photon. After rapid non-radiative relaxation, Pr$^{3+}$ decays from the $^3P_0$ level to lower-lying levels, emitting a green or red photon. (**b**) Schematic of the experiment used to detect photon bunching. The photon cutting material is excited at 180 nm using a ultraviolet lamp with a 180 ± 10 nm bandpass (BP) filter. The cascade emission is collected with a high numerical aperture objective, after which other emission lines and background in the ultraviolet are filtered out using a 350 nm longpass (LP) filter. The two emissions of the cascade process are spectrally separated using a dichroic (DC) filter and sent to two independent detectors. (**c**) Spectrum of the emission reflected into detector 1 (blue) compared with the unseparated emission spectrum (dashed). (**d**) Spectrum of the emission transmitted to detector 2 (red) compared with the unseparated emission spectrum (dashed). (**e**) Measured cross-correlation $g^{(2)}$ between the detection events on detectors 1 and 2 (see **b**) for photon-cutting emission from NaLaF$_4$:Pr$^{3+}$. Inset: comparison of the photon bunching signal $g^{(2)} - 1$ (green) with the photoluminescence decay curve of the $^3P_0$ level (red), on a semi-logarithmic scale.

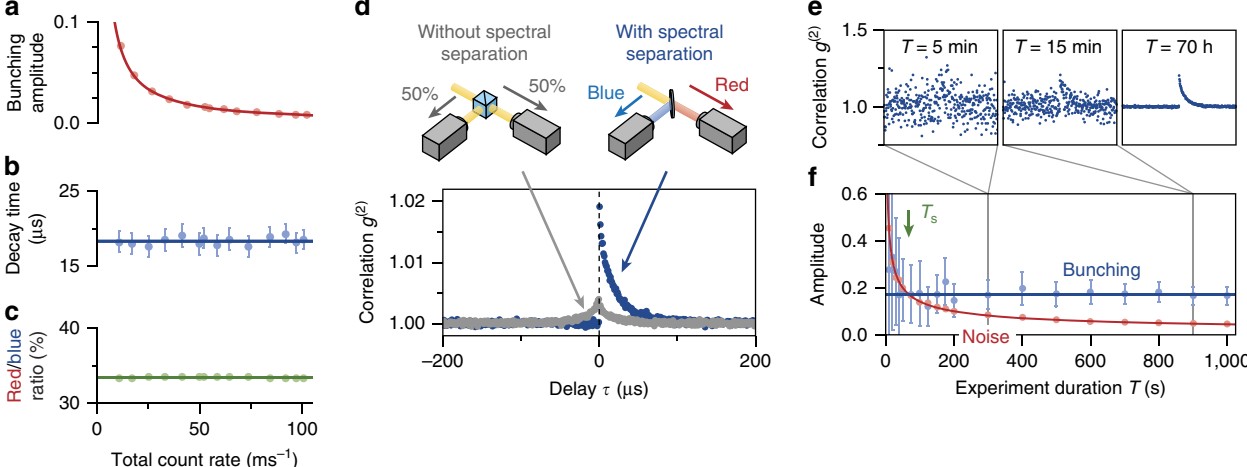

**Figure 3 | The effects of experimental parameters and sample properties.** (**a**) The photon bunching amplitude in NaLaF$_4$:Pr$^{3+}$ depends on the excitation density $\Phi$ (dots). The red line is a fit to equation (2). (**b**) Decay times of the bunching signal at various excitation densities. The horizontal line is the average of 18.4 μs. The error bars are the $2\sigma$ confidence interval of the fit results. (**c**) Ratio between the count rates at detectors 2 and 1 (dots). The horizontal line is the average ratio of 33%. (**d**) Comparison between a photon bunching experiment with spectral separation (blue) and without spectral separation (grey) under otherwise equal conditions. (**e**) Cross-correlation function for experiments with spectral separation, of varying experiment duration $T$. Longer experiments result in lower noise. (**f**) Noise amplitude (red) and bunching amplitude (blue) as a function of experiment duration $T$. The noise scales with $1/\sqrt{T}$, as indicated with the fitted line (red). The bunching amplitude does not depend on $T$, but a larger $T$ results in lower noise and a smaller uncertainty in the bunching amplitude. The error bars are $2\sigma$ confidence intervals on the fitted bunching amplitudes.

this we can estimate that NaLaF$_4$:Pr$^{3+}$ emits 0.35 red photons per 1 blue photon. This agrees well with the value of 0.40 reported by Herden et al.[28] and with the value of 0.33 obtained from the ratio of the count rates (Fig. 3c).

We have shown that the bunching amplitude equals $1/\bar{N}_X$, where $\bar{N}_X$ is the steady-state population in intermediate state X. Throughout this work, we used low excitation densities, so that $\bar{N}_X = 5$–100 (Figs 2e and 3) in our bulk powder. Such excitation

is strong enough that the photon count rate is reasonable ($10^3$–$10^4$ s$^{-1}$), but weak enough for a significant bunching amplitude. In Fig. 3e,f we investigate how long a measurement using weak excitation ($\bar{N}_X = 5$; total count rate 1,500 s$^{-1}$) must last to clearly observe bunching over the Poissonian background. Figure 3e shows correlation functions for experiments with different durations $T$. In Fig. 3f we plot the fitted bunching amplitude as a function of experiment duration, including the $2\sigma$

confidence interval on the fit (blue) and the standard deviation of the noise (red). Already after 5 min, the bunching amplitude exceeds the noise by $2\sigma$ and after 15 min by $6\sigma$, although photon bunching is not yet clear by visual inspection of the cross-correlation function (Fig. 3e). The time $T_s$ at which the bunching amplitude and noise level are equal (green arrow in Fig. 3f) can be used to calculate the detection efficiency $\eta$ if either the efficiency of the first or second radiative transition is known (see Supplementary Information equation (16)). Based on the efficiencies reported by Herden et al.[28], we estimate that our detection efficiency is $\eta = 0.4\%$.

## Discussion

Our work has demonstrated theoretically and experimentally that a photon-cutting material exhibits bunched emission, even if it contains a macroscopic number of optical centres. In fact, the magnitude of bunching does not depend on the total number of optical centres, but only on the steady-state population of centres in the excited state. We predict that similar photon statistics should be observable for many other photon-cutters recently reported[3,4,6,15–17,20–23,26,27]. An interesting analogy exists with bunched cathodoluminescence, which has recently provided evidence that the impact of an individual electron on a semiconductor material can generate multiple excitations[31,32].

Photon-bunching experiments will provide definite proof of photon cutting for materials where controversy exists[20,24]. The magnitude of the bunching signal depends on the photon-cutting efficiency of the material, as well as on the detection efficiencies (see Supplementary Note 1). With careful calibration of the experimental setup and reference measurements on one of the steps of the cascade, it is possible to determine the absolute photon-cutting efficiency of materials. We envision that experiments as presented here will contribute to the development and optimization of existing and new photon-cutting materials.

## Methods

**Photon correlation.** We used a microcrystalline NaLaF$_4$:Pr$^{3+}$ 1% powder provided by Jüstel and colleagues[28]. Approximately 10 mg of powder was glued as a thin layer of a few mm$^2$ to a non-luminescent background using SPI silver paint. The spectral output of a Micropack DH-2000 deuterium lamp filtered using an Acton Optics & Coatings 180-N 180 ± 10 nm bandpass filter illuminated the sample homogeneously, exciting Pr$^{3+}$ to a 4f$^1$5d$^1$ level, from which rapid non-radiative relaxation to the $^1S_0$ level takes place. The excitation intensity was controlled with two apertures of adjustable size. A Zeiss LD EC-Epiplan-NEOFLUAR 100 × numerical aperture 0.75 objective collected the emission light and emission not originating from the cascade pathway was filtered out with a 350 nm longpass filter. Two nominally identical Hamamatsu R10699 photomultiplier tubes, operated at 1,000 V and cooled with Peltier elements, were used as detectors. They have dark count rates of 10–20 cps. Becker & Hickl GmbH HFAC-26dB amplifiers were used to amplify the signals, which were recorded with a PicoQuant TimeHarp 260 photon counting module, operated in time-tagged time-resolved mode and set at a discriminator level of − 100 mV. In Supplementary Fig. 4 and Supplementary Note 4 we show that an experiment with one detector is not possible, because detector afterpulsing interferes with the detection of photon pairs. For the experiment with spectral separation (Figs 2e and 3), the emitted light was split with a Thorlabs DMLP425 dichroic mirror and sent to the two separate detectors. For the experiment without spectral separation (Fig. 3d, grey), the emission was divided equally over the two detectors using a Thorlabs BSW26 50/50 non-polarizing beamsplitter. Supplementary Note 3 describes how the raw list of photon arrival times is converted into the normalized cross-correlation function.

**Spectra.** Emission spectra were recorded on an Edinburgh Instruments FLS920 spectrofluorometer, using a Micropack DH-2000 deuterium lamp plus Acton Optics & Coatings 180-N 180 ± 10 nm bandpass filter for excitation and scanning a single monochromator after a 350 nm longpass filter to direct the cascade emission to a Peltier-cooled Hamamatsu R928P photomultiplier tube. The spectrum of the red-green photons was measured with the Thorlabs DMLP425 dichroic mirror in the emission path and the spectrum of the blue photons was obtained by subtracting the red-green spectrum from the spectrum in the absence of the

dichroic mirror. The luminescence decay curve of the $^3P_0$ was obtained by pulsed excitation with an Ekspla NT 342B laser at 446 nm and detection at 609 nm using a Triax 550 monochromator and Hamamatsu H7422-02 photomultiplier tube, coupled to a PicoQuant TimeHarp 260 photo counting module set at a discriminator level of − 100 mV.

**Derivation of bunching strength.** We derive the cross-correlation function for photons emitted by a macroscopic photon-cutting material with $N$ optical centres. Each centre has the general energy-level structure as in Fig. 1a: a highly excited state Y can decay to an intermediate excited state X and then further to the ground state G. The two steps have total transition rates $k_{YX}$ and $k_{XG}$, respectively, and radiative efficiencies $\eta_{YX} = k^r_{YX}/k_{YX}$ and $\eta_{XG} = k^r_{XG}/k_{XG}$ with the superscript 'r' denoting the radiative part of the transition rate. A continuous-wave light source pumps the centres from state G to state Y at an excitation rate $\Phi = \sigma I/\hbar\omega$, with $\sigma$ the absorption cross-section, $I$ the pump intensity and $\hbar\omega$ the photon energy. The two photons emitted in the cascade process are separated spectrally and sent to two detectors (1 and 2, respectively), each with zero dark count rate. This model describes our experiment on Pr$^{3+}$ well, as explained in Supplementary Note 1. There we also discuss situations where spectral separation of the two emitted photons is not possible (such as for multi-exciton generation in colloidal quantum dots) or where dark counts are not negligible.

A photon count on detector 1 signifies that one of the optical centres in the material just underwent the transition Y → X. This particular centre is therefore in the intermediate excited state X directly after the detection event and can subsequently decay further to the ground state at a rate $k_{XG}$. The expectation value for the population of state X as a function of delay time $\tau$ after the click on detector 1 is

$$\langle N_X(\tau) \rangle = \bar{N}_X + e^{-k_{XG}\tau}, \qquad (3)$$

with $\bar{N}_X = \Phi N/k_{XG}$ the steady-state population of state X. The decay of this one centre has a negligible effect on the excited-state populations $\bar{N}_X$ and $\bar{N}_Y = \Phi N/k_{YX}$ in the rest of the material, because we consider a macroscopic number of optical centres $N \gg 1$. Realizing that the detector count rates ($I_1$ and $I_2$) are proportional to the excited-state populations, we can express the normalized cross-correlation of the signals on detectors 1 and 2 as:

$$g^{(2)}(\tau) = \frac{\langle I_1(t)I_2(t+\tau) \rangle}{\langle I_1(t) \rangle \langle I_2(t) \rangle} = \frac{\bar{N}_Y \langle N_X(\tau) \rangle}{\bar{N}_Y \bar{N}_X}$$

$$= 1 + \frac{1}{\bar{N}_X} e^{-k_{XG}\tau} \qquad (4)$$

**Monte Carlo simulation.** We performed rejection-free kinetic Monte Carlo simulations of an ensemble of photon-cutting optical centres, each with an energy-level structure as in Fig. 1a. The simulation keeps track of the populations $N_G$, $N_X$ and $N_Y$ in the ground state G, the intermediate state X and the highest-energy excited state Y. Three processes can occur in the ensemble of optical centres: (1) absorption at rate $\Phi N_G$, (2) decay of a centre in state Y at rate $k_{YX}N_Y$ and (3) decay of a centre in state X at rate $k_{XG}N_X$. A simulation step consists of (A) randomly selecting one of the processes (1–3) to occur, taking into account the relative probabilities, (B) drawing a residence time $\Delta t$ from an exponential distribution with average $(\Phi N_G + k_{XG}N_X + k_{YX}N_Y)^{-1}$, (C) updating the simulation time to $t \to t + \Delta t$, (D) storing time $t$ in case of photon emission and (E) updating the populations of the three states. In the simulation of experiments with spectral separation, photon emission events from states X and Y are stored in separate channels. In the simulation of experiments with finite detector efficiency, photon emission events are randomly excluded from the storage. Dark count events were added separately by drawing the intervals between events from an exponential distribution with average $D^{-1}$, where $D$ is the dark count rate.

**Data availability.** The data that support the findings of this study are available from the corresponding author upon reasonable request.

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

## Acknowledgements

We are grateful to Thomas Jüstel and co-workers for supplying us with the $NaLaF_4:Pr^{3+}$ 1% sample. The work was supported by the EU Marie Curie Initial Training Network LUMINET (316906) and by the Netherlands Center for Multiscale Catalytic Energy Conversion (MCEC), an NWO Gravitation programme funded by the Ministry of Education, Culture and Science of the government of the Netherlands.

## Author contributions

F.T.R. conceived the idea of the experiment. M.d.J. designed, performed and analysed the experiment. F.T.R. derived the analytical correlation functions. M.d.J. performed the Monte Carlo simulations. M.d.J., A.M. and F.T.R. discussed the results and wrote the manuscript.

## Additional information

**Competing interests:** The authors declare no competing financial interests.

**Publisher's note**: 

