## [Peer Review File · Nature Communications]

Reviewers' comments:

Reviewer #1 (Remarks to the Author):

The submitted paper propose to use time correlation spectroscopy to demonstrate without any ambiguity the occurrence of quantum cutting effect in NaLaF₄:Pr³⁺. Indeed, former works in that field always tend to prove the effect using comparison of excitation spectra showing then that quantum yield can be over 1. Nevertheless excitation spectra are always experimental conditions rendering comparison hardly fully reliable. Since quantum cutting corresponds to the successive cascade emission of 2 photons from a single emitter, the author are using start-stop measurement where the first photon from the cascade is the triggering the electronics (start) while the second photon down to the ground state is the stop. This approach enables thus, to measure the decay time of the intermediate level and to demonstrate that the 2 photons arise from the same emitter. It corresponds to the bunching technics. Playing with 2 experimental conditions, with spectral separation and without spectral selection on the 2 detections channel, the authors achieve for the first time to estimate the yield of the quantum cutting effect which is, beyond the demonstration of the correlation between red/blue photon, the main result.

I consider that the experimental evidence is clear and that this paper brings new insights in the field of quantum cutting . I nevertheless invite the authors to revise their manunscript for the following reasons. The Monte Carlo simulation is, from my point of view useless and bring more confusion than help for the understanding. Indeed, bunching technic is rather classical, and eq 2 is well admitted. In addition, the population eq can be solved without MC. Last remark for Fig1-g, what does mean N=1 when keeping Nx=2?? Why the simulation is diverging from the analytical solution. It seems even that that it does not reproduce the curve correctly for N=10. I suggest to entirely remove the section on MC and the fig:1-g which does not bring additional information. For clarity, it would be better to use another sign for excitation rate, since X is also the label of the level.

When comparing the case of various Nx, the authors claim that the excitation has no effect on the signal to noise ration which is not clear. It suggests that Nx has no effect on the results. In such a situation, 2 goals might be forseen. Either, the experiment tends to demonstrate the occurrence of the quantum-cutting effect, or the goal is to extract parameter from the g₂ fitting. In the latter case, I am not sure that the confidence on the deduced k_x is the same for all the Nx. A small contrast with a high S/N ration might not be equivalent to a high contrast with a weak S/N. I suggest to improve the statistical analysis from that respect.

The information on N about 10¹⁷-10¹⁸ / 10mg is useless since the spot size is not given. The conclusion is confusing: "macroscopic number of optical center...", having a large N is not supposed to kill the bunching effect, it can only add a background of uncorrelated photons. Nx is supposed to change the contrast, not N.

Reviewer #2 (Remarks to the Author):

report on "Non-Poissonian photon statistics from macroscopic photon cutting materials" by de Jong et al.

The authors report on the theoretical and experimental evidence that photon-cutting materials have a photon emission statistics that is non-poissonian. More precisely, they prove that the correlation function between photons emitted from different energy level of the photon-cutting material exhibit a bunching-like curve.

The paper is very well written, the experiments and the theory are sound and convincing, the subject is exciting and quite timely, and last but not least, the findings are new and original. I can't but warmly recommend the paper for publication.

I have two small remarks; the first must be addressed, the second is left to the authors thinking:

1. I could not find the procedure for normalizing the g_2 . it is important to give it, and possibly justify it if this is not a fully coherent normalization (ie, using a laser as an input of their HBT interferometer)

2. It turns out that a very similar effect has been recently observed (Meuret et al., PRL, (2015)) in a quite different situation, namely intensity interferometry of the photons produced by an electron beam in interaction with a solid (cathodoluminescence). Although the physics rationale is quite different (1 electron create 1 plasmon that is able to excite several individual emitters in synchronization; therefore, the bunching is observed for any material), the maths seem very similar. However, in these experiments, the bunching amplitude is much higher than what observed by the authors, which is puzzling. Finally, this effect is used to measure the lifetime of several materials (see Meuret et al., ACS Photonics, 2016). In cathodoluminescence, lifetimes measurements are complicated, therefore this technique is useful. A possible application of the authors finding is a novel method to measure lifetime, *if* this is competitive with time resolved PL.

Our response to the reviewer report of Feb 7, 2017 on manuscript NCOMMS-16-27637-T:

We thank both Reviewers for carefully reading our manuscript and their constructive comments, which have been helpful to improve the manuscript. Below we reproduce the Reviewers' comments in black italic, and give our point-by-point response in blue. Changes to the manuscript based on the Reviewers' comments and suggestions are highlighted in red, both in this document and in the "Marked Manuscript" file attached.

.....

Response to Reviewer #1

The submitted paper propose to use time correlation spectroscopy to demonstrate without any ambiguity the occurrence of quantum cutting effect in NaLaF4:Pr3+. Indeed, former works in that field always tend to prove the effect using comparison of excitation spectra showing then that quantum yield can be over 1. Nevertheless excitation spectra are always experimental conditions rendering comparison hardly fully reliable. Since quantum cutting corresponds to the successive cascade emission of 2 photons from a single emitter, the author are using start-stop measurement where the first photon from the cascade is the triggering the electronics (start) while the second photon down to the ground state is the stop. This approach enables thus, to measure the decay time of the intermediate level and to demonstrate that the 2 photons arise from the same emitter. It corresponds to the bunching technics. Playing with 2 experimental conditions, with spectral separation and without spectral selection on the 2 detections channel, the authors achieve for the first time to estimate the yield of the quantum cutting effect which is, beyond the demonstration of the correlation between red/blue photon, the main result. I consider that the experimental evidence is clear and that this paper brings new insights in the field of quantum cutting . I nevertheless invite the authors to revise their manunuscript for the following reasons.

Our response: We are pleased to read that the reviewer recognizes the novelty and importance of our work.

The Monte Carlo simulation is, from my point of view useless and bring more confusion than help for the understanding. Indeed, bunching technic is rather classical, and eq 2 is well admitted. In addition, the population eq can be solved without MC. Last remark for Fig1-g, what does mean N=1 when keeping Nx=2?? Why the simulation is diverging from the analytical solution. It seems even that that it does not reproduce the curve correctly for N=10. I suggest to entirely remove the section on MC and the fig:1-g which does not bring additional information.

Our response: We apologise that our discussion of the Monte Carlo simulations was too concise. The Reviewer correctly points out that this could lead to confusion about (i) the deviations between the Monte Carlo simulation and the analytical equation (Eq. 2) and about (ii) the meaning of a constant steady-state population of $\bar{N}_x = 2$.

As mentioned by the Reviewer, Figure 1g shows differences between the analytical and the Monte Carlo model at $N = 1$ and $N = 10$. These arise because the analytical model neglects depletion of the ground state of optical centres. This is a good approximation for macroscopic N at low excitation rate, but not for a small number of centres. The Monte Carlo simulation thus validates the analytical model in the limit of large N , but also shows its break-down at small N .

Action 1: On page 2 of the revised manuscript we clarify what we mean by a constant steady-state population \bar{N}_X of 2:

“The excitation rate is set at $\Phi = 2k_{XG}/N$, which corresponds to a constant steady-state population of $\bar{N}_X = 2$ in the limit of large N .”

And a few lines down we discuss the deviations between analytical and Monte Carlo model:

“The analytical model (equation (2)) matches well with the Monte Carlo results for an ensemble of emitters with $N > 10$, including macroscopic materials containing a number of optical centres on the order of Avogadro's constant (Fig. 1g, green and blue), clearly revealing the occurrence of photon-pair emission in the photon statistics. The analytical model is less accurate for a small N (yellow and red), because the approximation of no ground-state depletion is justified only for large N and low excitation rates.”

For clarity, it would be better to use another sign for excitation rate, since X is also the label of the level.

Our response: This is a good suggestion of the Reviewer.

Action 2: We use “ Φ ” as the symbol for excitation rate throughout the revised manuscript and the Supplementary Information.

When comparing the case of various N_x , the authors claim that the excitation has no effect on the signal to noise ration which is not clear. It suggests that N_x has no effect on the results. In such a situation, 2 goals might be forseen. Either, the experiment tends to demonstrate the occurrence of the quantum-cutting effect, or the goal is to extract parameter from the g_2 fitting. In the latter case, I am not sure that the confidence on the deduced k_x is the same for all the N_x . A small contrast with a high S/N ration might not be equivalent to a high contrast with a weak S/N. I suggest to improve the statistical analysis from that respect.

Our response: We thank the Reviewer for suggesting that we discuss the effect of excitation density on the experiment and analysis in more detail. It is indeed counterintuitive that the bunching amplitude and noise should have the same scaling with excitation density. However, the theoretical model, which is firmly confirmed by the experiments (see Figs. 3a and 3f), explains this well. Based on the Reviewer's suggestion we have Monte Carlo simulated photon-bunching experiments for a wide range of excitation densities. Simple inspection of the simulated correlation functions illustrates that bunching amplitude and noise scale in the same way (see Supplementary Figure 3a–c, reproduced below). In fact, the simulated correlation functions are indistinguishable except for the range on the y-axis. Statistical analysis of the fit results then shows that the uncertainty in extracting the excited-state lifetime τ or bunching amplitude A is constant with excitation density, except at very low density where assumptions of the weighted least-squares fit procedure are no longer strictly valid.

Action 3: We included a new figure in the Supplementary Information where we investigate the effect of excitation density on the signal and noise in more detail:

Supplementary Figure 3 | The effect of excitation density on bunching amplitude and noise. (a–c) Simulated cross-correlation functions for photon-cutting emission by $\text{NaLaF}_4:\text{Pr}^{3+}$, **(b)** for the experimental parameters as in Fig. 2 of the main text, **(a)** for 10× weaker excitation, and **(c)** for 1000× stronger excitation. The bunching amplitude and noise both scale inversely with excitation power, so that the three correlation functions can be distinguished only from the range depicted on the y-axis. **(d,e)** From a statistical analysis of 1000 simulated experiments per excitation density, we calculate how accurately we can extract **(d)** the excited-state lifetime and **(e)** the bunching amplitude from a weighted least-squares fit. We consider a range of four orders of magnitude in excitation density normalised to the density of used in Fig. 2 of the main text. Open circles depict the variation of fitted values from 1000 simulations (as one standard deviation), filled circles the average fit uncertainty (as one standard error) of the 1000 fits. The different colors represent different experiment durations of $T = 7$ h (blue), $T = 70$ h (green; as in Fig. 2 of the main text), and $T = 700$ h (red). While longer experiments allow for more accurate estimates of the excited-state lifetime and bunching amplitude, the excitation density has only limited effect except at very low densities. For such low densities the uncertainty is slightly increased. This is the result of the assumption of Gaussian noise in the weighted least-squares fit procedure used, and could be potentially be avoided by using more advanced models for the noise [Bajzer1991].

[Bajzer1991] Bajzer, Z., Therneau, T. M., Sharp, J. C. & Prendergast, F. G. Maximum likelihood method for the analysis of time-resolved fluorescence decay curves, *Eur. Biophys. J.* **20**, 247–262 (1991)

On page 2 of the main text we refer to this new Supplementary Figure 3:

“See Supplementary Figure 3 for an analysis of what this means for the fit uncertainties.”

The information on N about 10^{17} - 10^{18} / 10mg is useless since the spot size is not given.

Our response: Providing this approximate number illustrates that the experiment is performed on an enormous number of optical centres simultaneously. The deep-UV illumination was attenuated using apertures of adjustable size (as depicted in Fig. 2b) and afterward not focused on the sample. This creates a homogeneous illumination, ensuring that all optical centres in the powder are addressed.

Action 4: On page 3 of the revised manuscript we write:

“We use ~10 mg of the material, containing approximately $N = 10^{17} - 10^{18}$ optical centres that in the experiment are excited more or less homogeneously.”

And we revised the Methods section to read:

“~10 mg of powder was glued as a thin layer of a few mm² to a non-luminescent background using SPI silver paint. The spectral output of a Micropack DH-2000 deuterium lamp filtered using an Acton Optics & Coatings 180-N 180±10 nm bandpass filter illuminated the sample homogeneously, exciting Pr³⁺ to a 4f¹5d¹ level, from which rapid non-radiative relaxation to the ¹S₀ level takes place.”

The conclusion is confusing: “macroscopic number of optical center...”, having a large N is not supposed to kill the bunching effect, it can only add a background of uncorrelated photons. Nx is supposed to change the contrast, not N.

Our response: The Reviewer summarises one of our main findings very well. This should be mentioned clearly in the concluding paragraph.

Action 5: On page 4 of the revised manuscript we added:

“In fact, the magnitude of bunching does not depend on the total number of optical centres, but only on the steady-state population of centres in the excited state.”

.....

Response to Reviewer #2

The authors report on the theoretical and experimental evidence that photon-cutting materials have a photon emission statistics that is non-poissonian. More precisely, they prove that the correlation function between photons emitted from different energy level of the photon-cutting material exhibit a bunching-like curve. The paper is very well written, the experiments and the theory are sound and convincing, the subject is exciting and quite timely, and last but not least, the findings are new and original. I can't but warmly recommend the paper for publication. I have two small remarks; the first must be addressed, the second is left to the authors thinking:

Our response: We thank the Reviewer very much for his/her compliments and the positive recommendation.

1. I could not find the procedure for normalizing the g2. it is important to give it, and possibly justify it if this is not a fully coherent normalization (ie, using a laser as an input of their HBT interferometer)

Our response: We normalise the cross-correlation function $g^{(2)}$ from the discrete raw data (the time tags of individual photon counts) using the average intensities on the two detectors, as according to equation (4). Indeed, this procedure yields correlation functions that approach unity at long delay times, and at negative delay times in the case of spectral separation (Figs. 2e, 3d, 3e). We agree with the Reviewer that it is important to clearly explain the procedure.

Action 6: In the revised Methods section we write:

“Supplementary Discussion 3 describes how the raw list of photon arrival times is converted into the normalised cross-correlation function.”

And the Supplementary Information (page 6) we now write:

“We normalise the cross-correlation function by multiplying with $(I_1 I_2 T \delta \tau)^{-1}$, where $I_{1,2}$ are the average intensities on detectors 1 and 2.”

2. It turns out that a very similar effect has been recently observed (Meuret et al., PRL, (2015)) in a quite different situation, namely intensity interferometry of the photons produced by an electron beam in interaction with a solid (cathodoluminescence). Although the physics rationale is quite different (1 electron create 1 plasmon that is able to excite several individual emitters in synchronization; therefore, the bunching is observed for any material), the maths seem very similar. However, in these experiments, the bunching amplitude is much higher than what observed by the authors, which is puzzling. Finally, this effect is used to measure the lifetime of several materials (see Meuret et al., ACS Photonics, 2016). In cathodoluminescence, lifetimes measurements are complicated, therefore this technique is useful. A possible application of the authors finding is a novel method to measure lifetime, *if* this is competitive with time resolved PL.

Our response: We thank the Reviewer for bringing to our attention this interesting analogy between bunching in cathodoluminescence (CL) and bunching by quantum cutting. We can hypothesise here that the lower bunching amplitude in our experiments compared to the CL experiments cited by the Reviewer might have two reasons: (i) our material emits at most 2 photons per absorption event, while in CL one high-energy electron can generate more photons; (2) the bunching amplitude is inversely proportional to the lifetime of the optical centres (Supplementary Information, equation (1)), which is much longer in our experiment (18 μ s) than in the references given by the Reviewer (up to 26 ns).

While for *cathodoluminescence* the analysis of the bunching statistics is a clever way to determine excited-state lifetimes, we would argue that for (photon-cutting) *photoluminescence* a direct measurement with a pulsed laser is simpler (see Fig. 2e). Nevertheless, based on the suggestion of this Reviewer as well as Reviewer #1, we analysed the uncertainties in fitting the excited-state lifetime of a photon cutter from the photon statistics in more detail in the new Supplementary Figure 3.

Action 7: In the “Discussion” paragraph of the revised manuscript (page 4) we write:

“An interesting analogy exists with bunched cathodoluminescence, which has recently provided evidence that the impact of an individual electron on a semiconductor material can generate multiple excitations [1,2].”

[1] Meuret, S. *et al.*, Photon bunching in cathodoluminescence, *Phys. Rev. Lett.* **114**, 197401 (2015)

[2] Meuret, S. *et al.*, Lifetime measurements well below the optical diffraction limit, *ACS Photonics* **3**, 1157–1163 (2016)

The excited-state lifetime of a photon cutter can be fitted from the correlation function of emitted photons. In the new Supplementary Figure 3, we analyse the fit uncertainties. See **Action 3** above.

REVIEWERS' COMMENTS:

Reviewer #1 (Remarks to the Author):

The authors addressed correctly the raised questions. The paper can be accepted.

Reviewer #2 (Remarks to the Author):

I was already warmly recommending publication in my last report; the authors having answered my minor comments, I am still warmly recommending publication

Our response to the reviewer report of Mar 17, 2017 on manuscript NCOMMS-16-27637A:

We are very grateful to both Reviewers for their fast and positive evaluation of our revised manuscript and, once again, for their useful suggestions in the first round of review.

.....

Reviewer #1:

The authors addressed correctly the raised questions. The paper can be accepted.

Reviewer #2:

I was already warmly recommending publication in my last report; the authors having answered my minor comments, I am still warmly recommending publication